# Antibiotics in Groundwater and River Water of Białka—A Pristine Mountain River

Anna Lenart-Boroń [1,*], Justyna Prajsnar [2], Maciej Guzik [2], Piotr Boroń [3], Bartłomiej Grad [3] and Mirosław Żelazny [4]

1 Department of Microbiology and Biomonitoring, Faculty of Agriculture and Economics, University of Agriculture, Mickiewicza Ave. 24/28, 30-059 Krakow, Poland
2 Jerzy Haber Institute of Catalysis and Surface Chemistry, Polish Academy of Sciences, Niezapominajek Str. 8, 30-239 Krakow, Poland
3 Department of Forest Ecosystems Protection, Faculty of Forestry, University of Agriculture, 29 Listopada Ave. 46, 31-425 Krakow, Poland
4 Department of Hydrology, Institute of Geography and Spatial Management, Jagiellonian University, Gronostajowa Str. 7, 30-387 Krakow, Poland
* Correspondence: anna.lenart-boron@urk.edu.pl

**Abstract:** Antibiotics are emerging pollutants of great concern, due to detrimental effects of their sublethal concentrations on microbial communities. Monitoring of antibiotics' presence and concentrations in the aquatic environment is of fundamental importance to the management of water resources. This study was aimed at filling the knowledge gap in terms of presence and concentration of antibiotics in surface water and groundwater in one of the mountain regions in southern Poland. The detailed aims included the assessment of whether there are spatial and/or temporal trends in antibiotic occurrence in water and the investigation of causes behind the changes in antibiotic concentrations. The study was conducted in seven sites (two groundwater and five river water) along the Białka river valley. Antibiotics were subjected to solid-phase extraction, followed by UHPLC/MS detection. Clindamycin, erythromycin, ofloxacin and trimethoprim were the most frequently detected, while the highest concentrations were observed for oxytetracycline and clindamycin. No antibiotics were detected in only one of the groundwater sites. Sewage treatment plant effluent was the hotspot of antibiotic contamination of the river downstream. The detection rates of antibiotics in the examined region seem to be driven mainly by the stability of antibiotics in the environment.

**Keywords:** antibiotics; aquatic environment; groundwater; mountain areas; sewage treatment plant; surface water

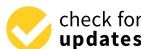



## 1. Introduction

Nowadays, it is impossible to imagine the functioning of humans in the modern world without chemicals, including antibiotic agents, which in the 20th century were considered a miracle discovery of medicine. However, they are excessively used in various branches of human activity; they are widely prescribed to treat bacterial infections, for prophylactic purposes in both human and veterinary medicine as well as growth-promoting agents in farm animals [1]. The main sources of antibiotics in the environment include: domestic misuse, agriculture and aquaculture, clinical waste and, recently, unjustified self-medication during the COVID-19 pandemic [2]. Because of their wide application for a number of purposes, antibiotics are continuously released into the environment. The metabolism rate of antibiotics in human and animal organisms is very low; therefore, about 25–75% of antibiotics enter the environment in the form of a parent compound [3]. Their persistence and mobility in the environment are high enough to allow them to penetrate from wastewater treatment plants, household and agricultural waste or manure used as natural fertilizers into surface water and groundwater. The use of antibiotics in farming

and ineffective elimination of these compounds by wastewater treatment plants are the main reasons for antibiotics' presence in the aquatic environment [4].

Antibiotics have been considered as emerging pollutants of greatest concern because of their potential detrimental effect on the environment, including their impact on accelerating the bacterial resistance to these compounds and the resulting implications on human health [5]. The concentrations of antibiotics in the aquatic environment are not high enough to directly threaten human health, but they exert selective pressure on the microbial community, affecting the occurrence of genetic mutations and promoting the antibiotic resistance transfer rate through vectors [3]. The fact that antibiotic-resistant bacteria from anthropogenic sources can mix with naturally occurring environmental strains in the aquatic environment provides a risk of the occurrence of new resistant strains [6]. Moreover, the detrimental effect of antibiotics on microbial communities includes the disappearance or inhibition of some microbial groups involved in key ecosystem functions. In such ways, they reduce microbial diversity, which is crucial in the maintenance of biological processes in various environments, including water [7].

The monitoring of antibiotics' presence and concentrations in the aquatic environment is of fundamental importance to the environmental management of water resources and their quality. There is a large gap in the data on the presence and concentration of antibiotics in the aquatic environment in Poland. In 2014, Gbylik-Sikorska et al. [8] examined the occurrence of veterinary antibiotics in fresh water, sediment and fish of rivers and lakes by using liquid chromatography–tandem mass spectrometry. However, they did not detect any of the examined 45 antimicrobial agents above the limit of quantification. The second study on the detection of antimicrobial agents in water [9] was conducted on six sites along the sewage-impacted Vistula river and aimed to determine the presence of 32 antimicrobial agents. Similarly, as in the study by Gbylik-Sikorska et al. [8], many substances were below the method detection limits or quantification limits. However, azithromycin, clarithromycin, fluconazole and sulfamethoxazole were detected in the concentrations exceeding 10 ng/L. Apart from these two studies, a paper by Lenart-Boroń et al. [10] presents the initial results of the antibiotic detection in three sites along the Białka river and their effect on the microbial community in water. A further study [11] focused on the effect of the COVID-19 pandemic's lockdown on a number of water-quality parameters, including antibiotic content. However, the examinations of antibiotic presence and content started in [10] and [11] go much further and deeper.

Given the importance of the above-described issue to environmental and public health and the scarcity of data, this study aimed at shedding some more light on the presence and concentration of antibiotics in surface water and groundwater in one of the mountain regions in southern Poland. More detailed aims of this study included: (i) to assess whether there are spatial and/or temporal trends in antibiotic occurrence in surface water and groundwater; (ii) to investigate the causes behind the changes in antibiotic concentrations in surface and groundwater.

## 2. Materials and Methods

### 2.1. Sample Collection

The samples of river water (RW) and groundwater (GW) were collected in seven sampling campaigns: March, August and December 2019 and January, February, March and May 2020. The sampling campaigns corresponded to both varying numbers of tourists visiting the area [12] and the expected seasonal changes in antibiotic consumption [13].

The studied area included five surface-water sampling points (RW) located along the mountain river Białka (southern Poland) and two groundwater (GW) sampling points (piezometers) located in the Tatra National Park, which directly neighbors the Białka river valley (Figure 1). The studied sites differ in terms of anthropogenic pressure put thereon, understood as the close or distant proximity of houses, hotels, ski stations and the related infrastructure or livestock-associated pressure.

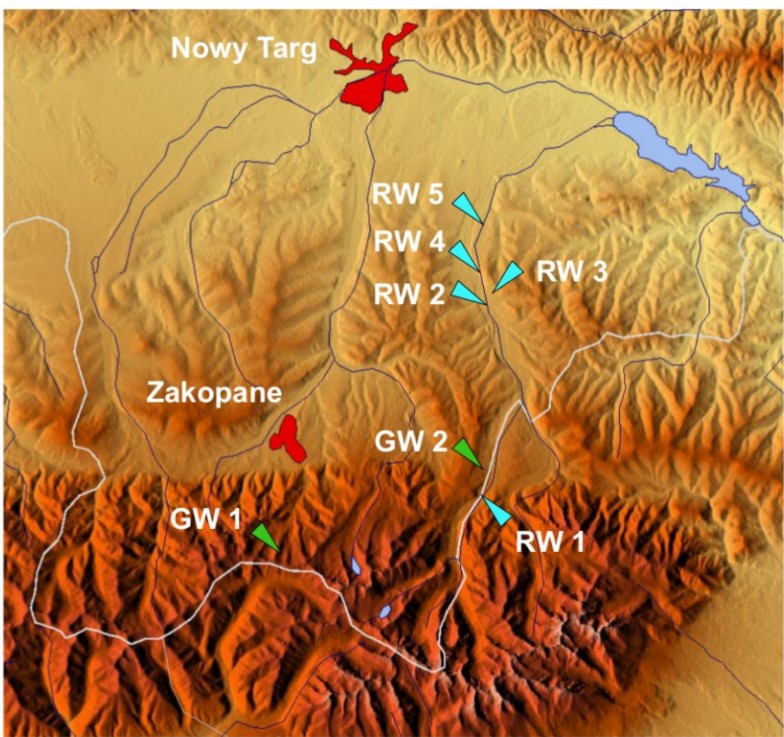

**Figure 1.** Study area and sampling sites.

Water temperature (T), electrolytic conductivity ($EC_{25\,°C}$) and pH were measured onsite with a handheld multimeter (YSI Pro 2030; USA, Yellow Springs, Ohio). For the determination of antibiotic content, 2l water samples were collected into non-transparent sterile polypropylene bottles, filled to overflowing to avoid possible UV, biological or oxygen degradation of antibiotics. If possible, the samples were processed as soon as they were transported to the laboratory. Otherwise, they were stored at −20 °C until analysis.

### 2.2. Reagents and Chemicals

All solvents used in this study were HPLC grade (99,8%, VWR International LLC, Gdańsk, Poland). Antibiotics were selected based on their wide application in human and veterinary medicine [14,15]. Water and methanol extracts of antimicrobial susceptibility disks were used for qualitative analysis (tigecycline (Oxoid, Basingstoke, UK), tylosin 30 μg (Liofilchem, Abruzzi, Italy), enrofloxacin 5 μg (Oxoid), cefotaxime 30 μg (Oxoid), cefazolin 30 μg (Oxoid), rifampicin 5 μg (Oxoid)). For the purpose of quantitative analysis, only pure standards of antibiotics were used and these were: oxytetracycline (VWR, Pennsylvania, USA), vancomycin (VWR), erythromycin (POL-AURA, Morąg Poland), trimethoprim (POL-AURA), ofloxacin (POL-AURA), sulfamethoxazole (POL-AURA), clindamycin (POL-AURA), cefoxitin (POL-AURA), ampicillin (Sigma-Aldrich), amoxicillin (Sigma-Aldrich, St. Louis, Missouri, US), netilmicin (POL-AURA), gentamicin (VWR), doxycycline (POL-AURA), tetracycline (POL-AURA), piperacillin (POL-AURA), ceftazidime (POL-AURA), cefuroxime (POL-AURA) and ciprofloxacin (POL-AURA)].

### 2.3. Analytical Procedure

Antibiotics present in water were subjected to solid-phase extraction (SPE) using Oasis HLB cartridges (6 cc Vac Cartridge, 500 mg Sorbent per Cartridge, 60 μm Particle Size, Waters). The SPE cartridges were conditioned using first 10 mL of methanol and then 5 mL of ultrapure water three times at a flow rate of 1–2 mL/min. Subsequently, the pre-filtered water samples (0.45 μm) were passed through the SPE cartridge (1 L each) at a flow rate of 10–20 mL/min. Next, the SPE cartridges were dried under vacuum for 30 min, and then

the compounds were eluted with 10 mL of methanol. Finally, the samples were completely dried and dissolved in 1 mL of methanol [16].

Then, qualitative and quantitative analysis of antibiotic content was conducted using an Agilent 1290 Infinity System UHPLC equipped with an autosampler and MS Agilent 6460 Triple Quad Detector (Santa Clara, CA, USA). For the separation of compounds, an Agilent Zorbax Eclipse Plus C18 column (2.1 × 50 mm, 1.8 µm) was used at 30 °C. The gradient of water with 0.1% formic acid and organic phase (acetonitrile with 0.1% formic acid) was applied: 0–5.50 min 5% organic phase, 5.51–8 min 100% organic phase, 8.01–9 min 95% organic phase in order to separate the compounds. The volume of the injected sample was 5 µL, and the flow rate was 0.4 mL/min. An MS Agilent 6460 Triple Quad tandem mass spectrometer with an Agilent Jet Stream ESI interface was used in both positive and negative ion polarization using Dynamic MRM mode. The optimum collision energy was variable depending on antibiotics [10]. Nitrogen was used as the drying gas and for collision-activated dissociation (flow rate 10 L/min). The temperature of drying gas and sheath gas was in both cases 350 °C. The capillary and nozzle voltages were set to 3500 V and 500 V, respectively.

### 2.4. Data Validation and Analysis

The effectiveness of the applied UHPLC/MS/MS method was verified by extracting a mix of all standards of antibiotics used for qualitative analysis. Calibration curves were obtained from pure compound standards. System control, data acquisition and data processing for UHPLC-MS was performed by MassHunter software (Version 10.0, Agilent, Santa Clara, CA, USA). The limits of detection (LOD) ranged from 0.083 to 83.3 ng/L, limits of quantification from 0.25 to 250 ng/L and SPE recovery ranged from 9.94% to c.a. 100% [10]. The varying SPE recovery rates resulted from their varying affinity to the SPE bed. The obtained SPE recovery values were used to calculate the final concentrations of antimicrobial agents in the water samples.

The normality of data distribution was tested using the Shapiro–Wilk test. The data were not normally distributed; therefore, non-parametric tests were applied in the further steps of statistical analysis. Basic descriptive statistics were calculated, and the significance of differences in the concentration of antimicrobial agents between the examined sites and years of analysis was determined using a Kruskal–Wallis test ($p < 0.05$). The tests were conducted in Statistica v. 13 (StatSoft, Tulsa, OK, USA). Heatmap construction, including clustering analyses, were conducted in R environment using pheatmap package [17].

### 3. Results

A total of 24 antimicrobial agents were monitored within this study, out of which 16 were detected above the LOD of the method (Table 1). The examined water samples differed in terms of the expected anthropogenic pressure, understood as the proximity to households, hotels, tourist infrastructure and livestock-associated pressure. Figure 2 presents the frequency of detection of antibiotics (only those above LOQ are shown in this figure) in the water samples, divided into three groups: groundwater (GW), river water (RW) and river water collected by the discharge from the sewage treatment plant (STP). The overall frequency of detection for all antimicrobial agents examined in this study is shown in Table 1. While looking at the overall frequency of detection, clindamycin was most frequently detected (63.16% of samples), followed by erythromycin (59.65%), ofloxacin and trimethoprim (57.89%) and sulfamethoxazole (50.88%). While looking at Figure 2, it can be noticed that these five antimicrobials are also the five most frequently detected in STP, RW and GW samples but in varying order. In descending order of detection frequency, these antimicrobials can be arranged as follows: STP—ofloxacin, trimethoprim (100%) > erythromycin, clindamycin (90%) > sulfamethoxazole (80%); RW—clindamycin (63%) > ofloxacin, erythromycin (53%) > trimethoprim (50%) > sulfamethoxazole (45%); GW—erythromycin (67%) > trimethoprim, sulfamethoxazole (50%) > clindamycin, ofloxacin (33%). On the other hand, gentamicin (aminoglycoside), amoxicillin, aztreonam, cefurox-

ime, cephazolin (β-lactams), tylosin (macrolide), rifampicin (ansamycin) and enrofloxacin (fluoroquinolone) were not detected at all in this study. The antimicrobial agents, the concentrations of which were higher than the LOD and/or LOQ of the UHPLC/MS/MS method but with the least frequently detected, belonged to the group of β-lactams and these were: ceftazidime and piperacillin (1.75%), cefamandole (3.51%) and ampicillin (5.26%).

As shown in Table 1, none of the examined antimicrobials were detected in one of the groundwater samples (GW1) collected in the Tatra Mountains (Tatra National Park). However, the second groundwater samples (GW2), also collected in the TNP but much closer to households and located below mountain shelters, was not as clean: as many as seven antimicrobial agents were detected in this sample: cefoxitin, erythromycin, ofloxacin, clindamycin, vancomycin, trimethoprim and sulfamethoxazole. The highest number of antimicrobial agents was detected by the STP (14 detected and 13 detected in the amounts exceeding the LOQ, which allowed precise assessment of their concentration). This sample was also characterized by the highest mean and maximum concentrations of all antimicrobials, except ceftazidime (the highest mean and max. concentration of this antibiotic were recorded in the RW4 sample, located c.a. 3 km downstream of the STP).

What can also be observed while looking at Table 1 is that the total concentrations of antimicrobial agents detected vary considerably, from 19.68 ng/L for ampicillin (β-lactam) to 6063.25 ng/L for oxytetracycline (tetracyclines). Oxytetracycline is also the antibiotic whose maximum concentration was the highest out of all examined antibiotics (1819.07 ng/L by the STP).

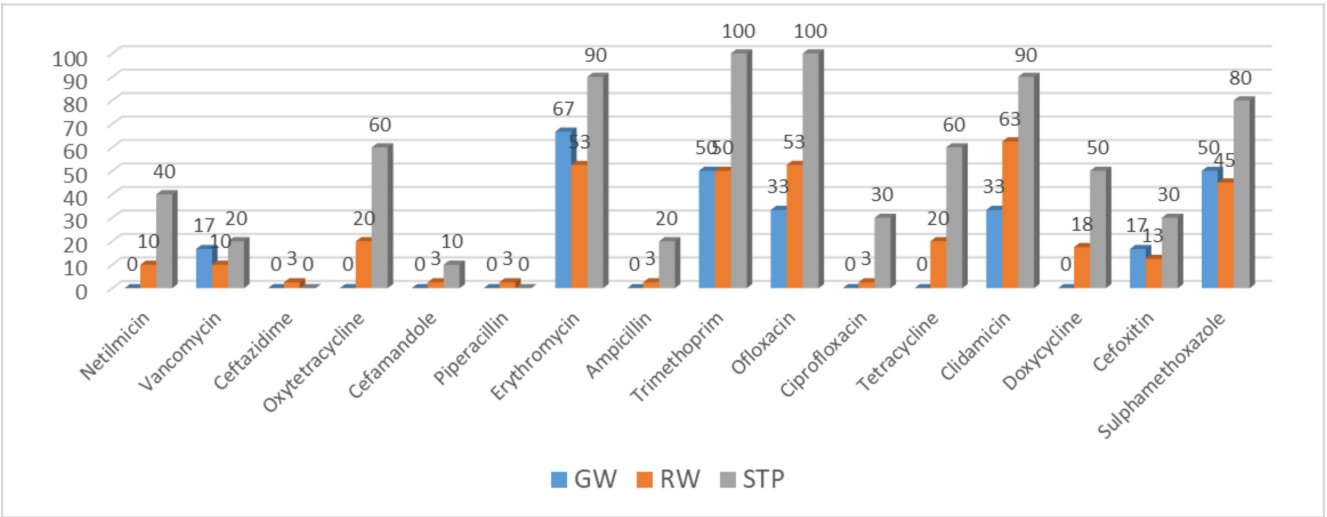

**Figure 2.** Frequency of antibiotic detection in various types of water samples.

Figure 3a–f show the differences in the total concentrations of antibiotics between the examined sites. The total concentration of antimicrobial agents varied from 17.94 ng/L for the river water upstream of the STP (RW2) to 7593.37 ng/L by the STP. The total concentration of antibiotics in groundwater (GW2) was 27.92 ng/L. Interestingly, the share of total concentration of vancomycin in all antimicrobials is very high in the GW and RW samples that are not subjected to high anthropogenic pressure—even 64.38% (17.91 ng/L) in GW and 53.31% (21.83 ng/L) in RW1 (Figure 3a–f). Oxytetracycline was not detected in GW, but starting from the RW1, its share in the total concentration of antimicrobials is also very high: from 7.92% (80.20 ng/L) in RW4 to as much as 79.32% (1829.50 ng/L) in RW5.

**Table 1.** Overall frequency of antibiotic detection (%) and concentrations of antimicrobial agents in the examined groundwater (GW) and river water (RW) samples. The values are presented as means of all measurements, values in brackets are min–max. The highest values of mean and max. concentrations, as well as the number and frequency of antimicrobials detected, are shown in bold. The concentration of antimicrobial agents marked with an * differ significantly between the examined sites ($p < 0.05$).

| Chemical Group | Antibiotic | Frequency of Detection (%) | GW 1 (Pristine) | GW2 (Pristine) | RW1 (Pristine) | RW2 (Low Impact) | RW3 (STP—High Impact) | RW4 (Moderate Impact) | RW5 (Moderate Impact) | Total Concentration of Antibiotic Detected (ng/L) |
|---|---|---|---|---|---|---|---|---|---|---|
| Aminoglycoside | Netilmicin | 14.04 | 0.0 | 0.0 | 0.0 (0–>LOQ) | 0.0 | **78.93 (0–424.89)** | 0.0 (0–>LOQ) | 34.57 (0–241.99) | 914.66 |
| | Gentamicin | 0.0 | 0.0 | 0.0 | 0.0 | 0.0 | 0.0 | 0.0 | 0.0 | 0.0 |
| B-lactam | Amoxicillin | 0.0 | 0.0 | 0.0 | 0.0 | 0.0 | 0.0 | 0.0 | 0.0 | 0.0 |
| | Ampicillin | 5.26 | 0.0 | 0.0 | 0.0 | 0.0 | **1.93 (0–11.57)** | 0.04 (0–0.39) | 0.0 | 19.68 |
| | Aztreonam | 0.0 | 0.0 | 0.0 | 0.0 | 0.0 | 0.0 | 0.0 | 0.0 | 0.0 |
| | Cefamandole | 3.51 | 0.0 | 0.0 | 0.0 (0–>LOQ) | 0.0 | 0.0 (0–>LOQ) | 0.0 | 0.0 | 0.0 |
| | Cefoxitin | 21.05 | 0.0 | 0.0 (0–>LOQ) | 0.0 (0–>LOQ) | 0.0 (0–>LOQ) | **96.55 (0–261.44)** | 2.33 (0–18.65) | 3.48 (0–17.39) | 808.40 |
| | Ceftazidime | 1.75 | 0.0 | 0.0 | 0.0 | 0.0 | 0.0 | **26.92 (0–296.10)** | 0.0 | 296.10 |
| | Cefuroxime | 0.0 | 0.0 | 0.0 | 0.0 | 0.0 | 0.0 | 0.0 | 0.0 | 0.0 |
| | Cephazolin | 0.0 | 0.0 | 0.0 | 0.0 | 0.0 | 0.0 | 0.0 | 0.0 | 0.0 |
| | Piperacillin | 1.75 | 0.0 | 0.0 | 0.0 | 0.0 | 0.0 | 0.0 | <LOD; >LOQ | 0.0 |
| Macrolide | Erythromycin * | 59.65 | 0.0 | 0.89 (0–3.38) | 0.42 (0–1.22) | 0.0 | **1.93 (0–5.68)** | 1.15 (0–7.45) | 0.12 (0–0.22) | 37.89 |
| | Tylosin | 0.0 | 0.0 | 0.0 | 0.0 | 0.0 | 0.0 | 0.0 | 0.0 | 0.0 |
| Ansamycin | Rifampicin | 0.0 | 0.0 | 0.0 | 0.0 | 0.0 | 0.0 | 0.0 | 0.0 | 0.0 |
| Fluoroquinolone | Ciprofloxacin | 7.02 | 0.0 | 0.0 | 0.0 (0–>LOQ) | 0.0 | **2.87 (0–14.26)** | 0.0 | 0.0 | 25.79 |
| | Enrofloxacin | 0.0 | 0.0 | 0.0 | 0.0 | 0.0 | 0.0 | 0.0 | 0.0 | 0.0 |
| | Ofloxacin * | 57.89 | 0.0 | 0.27 (0–1.35) | 0.0 (0–>LOQ) | 0.0 (0–>LOQ) | **8.32 (0.72–19.82)** | 2.82 (>LOQ–7.67) | 1.11 (>LOQ–2.39) | 115.50 |
| Tetracycline | Doxycycline * | 26.32 | 0.0 | 0.0 | 0.0 | 0.0 | **223.12 (0–1081.58)** | 35.75 (0–132.23) | 10.0 (0–59.99) | 604.87 |
| | Oxytetracycline * | 24.56 | 0.0 | 0.0 | 1.13 (0–9.14) | 0.67 (0–6.71) | 43.03 (0–**1819.07**) | 7.29 (0–38.28) | **291.62** (0–1749.67) | **6063.25** |
| | Tetracycline | 24.56 | 0.0 | 0.0 | 0.0 | 0.0 (0–>LOQ) | **2.62 (0–7.38)** | 0.80 (0–7.24) | 0.71 (0–2.25) | 32.43 |
| Lincosamid | Clindamycin * | 63.16 | 0.0 | 0.36 (0–1.82) | 0.17 (0–1.83) | 0.58 (0–2.1) | **25.65 (0–228.48)** | 20.36 (>LOQ–38.71) | 17.91 (>LOQ–24.01) | 1101.82 |
| Glycopeptide | Vancomycin | 12.28 | 0.0 | 2.99 (0–17.91) | 1.82 (0–21.83) | 0.64 (0–6.39) | **30.45 (0–153.71)** | 2.35 (0–25.81) | 2.24 (0–15.67) | 392.09 |
| Antifolate | Trimethoprim * | 57.89 | 0.0 | 0.29 (0–1.06) | 0.0 (0–>LOQ) | 0.02 (0–20) | **44.03 (0.20–171.00)** | 3.18 (>LOQ–6.21) | 1.65 (>LOQ–3.10) | 480.44 |
| Sulphonamide | Sulfamethoxazole * | 50.88 | 0.0 | 0.20 (0–0.82) | 0.0 (0–>LOQ) | 0.0 (0–>LOQ) | **8.62 (0–16.93)** | 2.00 (0–2.96) | 2.36 (0–5.71) | 106.65 |
| Number of antimicrobials detected/quantified | | | 0 | 7/6 | 11/4 | 8/4 | **14/13** | 13/12 | 12/11 | |

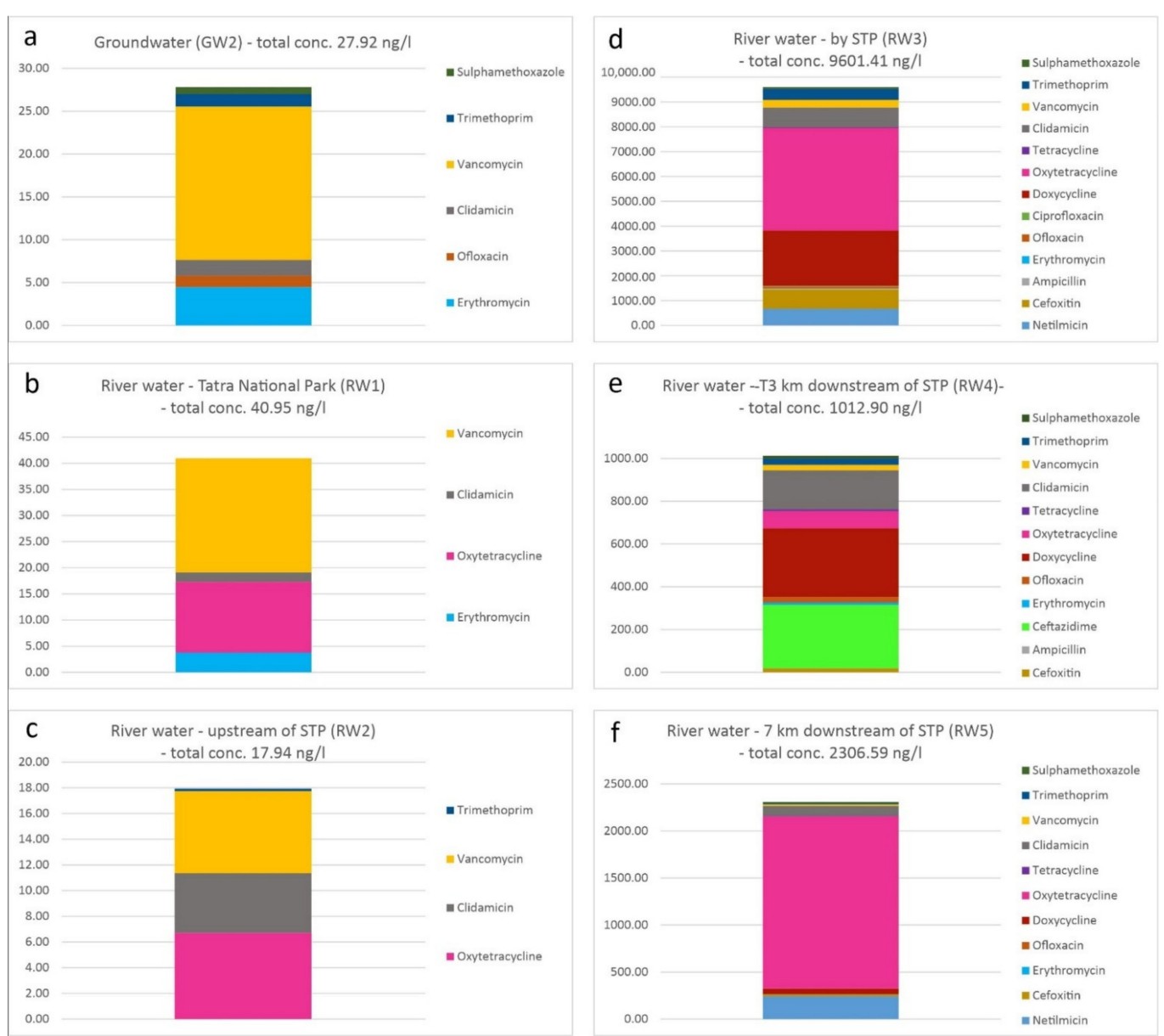

**Figure 3.** (**a**–**f**) Concentrations (ng/L) and share of antimicrobial agents in the total concentration of antimicrobial agents detected in each site.

Based on the log10+1 transformed data on the concentration of antimicrobial agents examined in this study, a heatmap was constructed showing the evident differences in the concentration of these compounds in the examined sites (Figure 4). The clustering of the examined sites clearly reflects the expected anthropogenic pressure put on the environment in the direct vicinity of the sampling sites. The site located by the STP(WWTP) clearly stands out from the remaining sites in terms of the highest concentrations of most of the detected antimicrobial agents. The sites described as subjected to moderate impact are RW3 and RW4, which are located c.a. 3 and c.a. 7 km downstream of the STP. The majority of antibiotics detected in this study were also present in these two sites but in smaller concentrations. The third cluster groups RW2 (located upstream of the STP and assessed as "low impact"), RW1 (river water collected at the Tatra National Park) and both groundwater samples. These four samples were assessed as "pristine", due to either very low or an absence of antimicrobials. The cluster analysis was also applied to the examined antimicrobial agents and shows four clusters. The first one is very distinct and groups clindamycin, doxycycline, netilmicin and oxytetracycline, which are antibiotics

whose concentrations are very high by the STP and still high in the "moderately impacted" RW4 and RW5 samples. The second cluster is also very distinct and groups ofloxacin, vancomycin, cefoxitin, trimethoprim and sulfamethoxazole, whose concentrations are high by the STP and much lower in the "moderately impacted" RW4 and RW5. The concentrations of antibiotics in the "low impacted" and "pristine" samples are almost negligible in terms of this analysis.

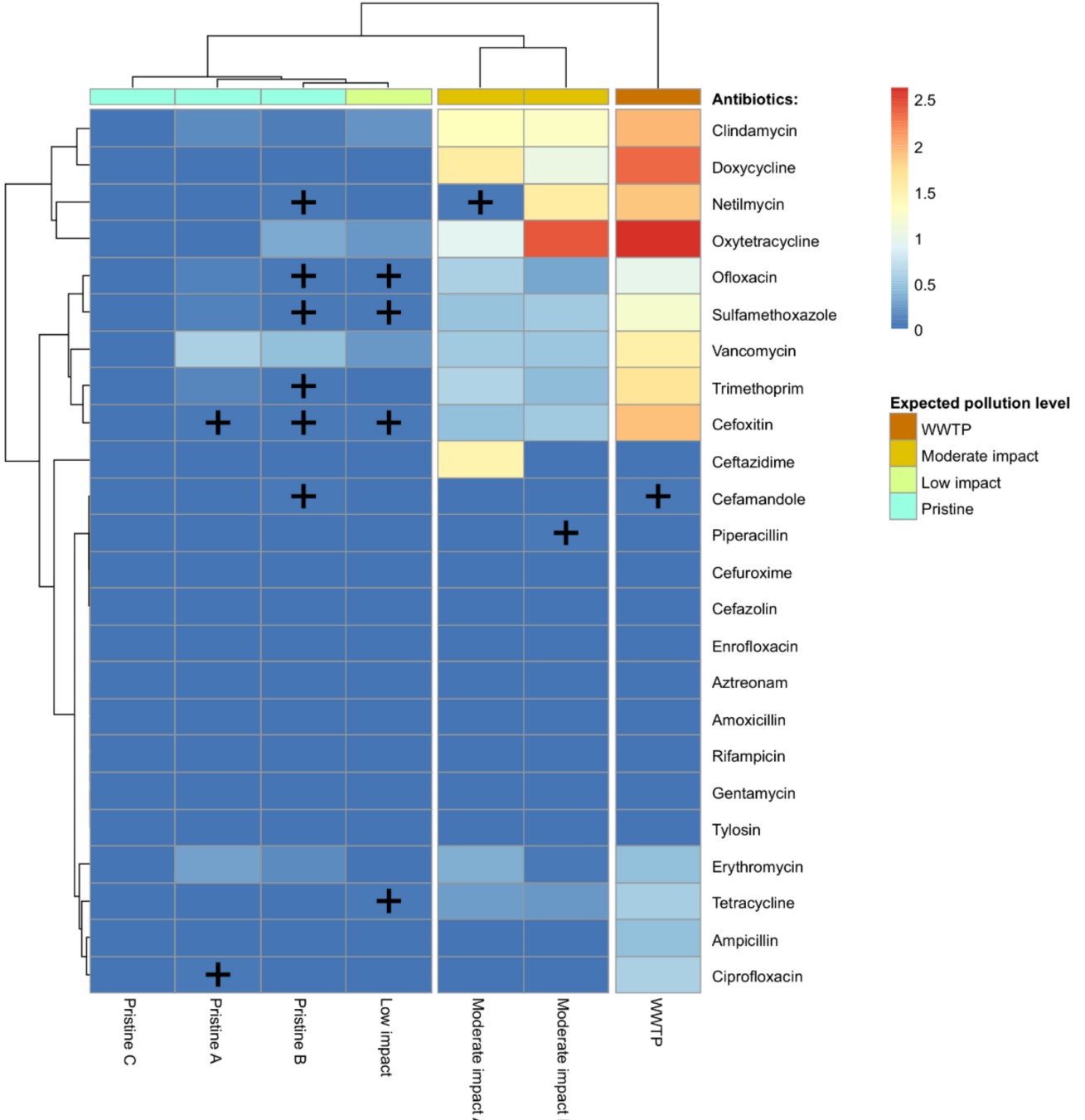

**Figure 4.** Antibiotic concentration heatmap constructed on the basis of log10+1 transformed data for 24 antimicrobial agents examined in this study, all dates combined. Please note the clear clustering of the sites corresponding to the expected anthropogenic pressure put on the water environment, reflected in the concentration of antibiotics. The "+" sign indicates the antimicrobial agents the presence of which was detected (i.e. exceeded the LOD) but did not exceed the LOQ value.

As part of this study, the concentrations of different groups of antibiotics detected in all of the examined samples and in the sample collected by the STP were compared to the annual human consumption of antibacterials for systemic use [15], as well as with sales data of veterinary antimicrobial agents [14], and presented in Figure 5. There are many discrepancies between the consumption and detection data. Firstly, there was much higher consumption of antimicrobial agents for systemic use reported to the ECDC in 2019 than in 2020, and the sales data of veterinary antibiotics are similar in both years studied. The trend of detection is reverse, as amounts of antimicrobials detected, both by the STP and in all samples cumulatively, are much higher in 2020 than in 2019. Tetracyclines were the antimicrobial agents whose detected concentrations were the highest among all tested groups, while their consumption for human systemic use is much lower (their concentration was fifth in 2019 while in 2020 it was nearly negligible (0.05 DDD). The sales of tetracyclines in veterinary medicine in 2019 and 2020 are the second in favor of penicillins. On the other hand, in human medicine, the consumption of penicillins was the highest in 2019, while in 2020 trimethoprim and sulfonamides were the most frequently consumed antimicrobials. Statistically significant differences (Kruskal–Wallis test) in the concentrations of detected antimicrobial agents between 2019 and 2020 were observed only for oxytetracycline (H = 12.66, $p = 0.0004$).

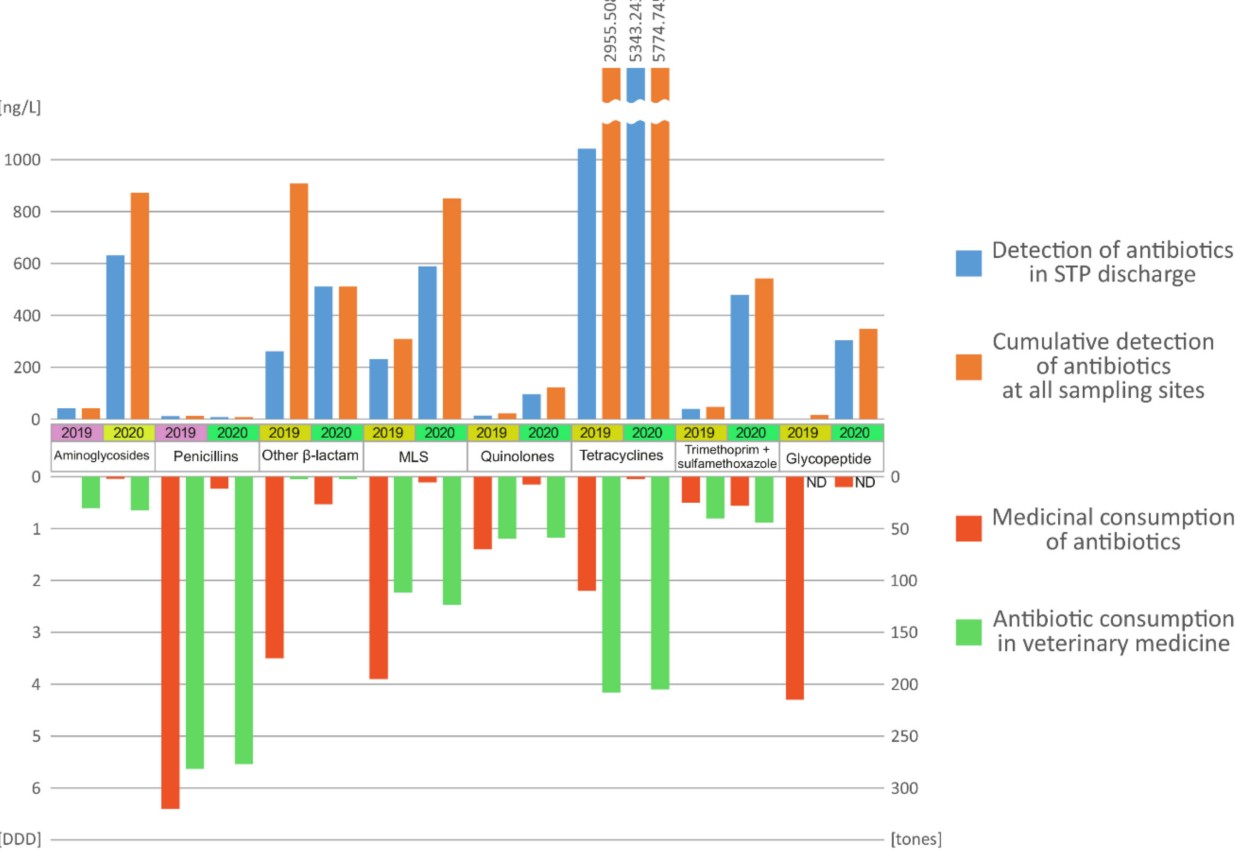

**Figure 5.** Detection of examined groups of antimicrobials at all sampling sites along with their detection by the STP (upper part of the graph) compared with their consumption in human and veterinary medicine (lower part of the graph).

## 4. Discussion

The presence of antimicrobial agents both in surface water and in groundwater is a serious problem, for a number of reasons. Threats related with water use for various purposes (including irrigation, as a source of drinking water and other urban, rural and environmental applications) and the associated toxicological implications on humans

and animals are among the most important ones [18]. Another main concern, associated with the release of antimicrobials into the environment, is related with their impact on the development of antibiotic resistance among bacteria and the resulting reduction of therapeutic potential of antibiotics [7,19].

In this study, the presence of 16 antimicrobial agents, classified into 10 groups, was detected in six out of seven samples, including one sample of groundwater in the pristine and protected area of the Tatra National Park. Generally, the frequency of detection and concentrations of antimicrobial agents in the aquatic environment are the result of factors such as: ubiquity of antibiotics administration for various purposes [19], their removal efficiency by the WWTPs [20] and their persistence in water, defined based on their half-life value [18]. In our study, clindamycin (lincosamides), erythromycin (macrolides), ofloxacin (fluoroquinolones), trimethoprim (antifolate) and sulfamethoxazole (sulfonamides) were most frequently detected. On the other hand, the highest concentrations were observed for oxytetracycline (tetracyclines), followed by clindamycin and netilmicin.

According to [20], three out of these seven antimicrobial agents were among the ones with the highest sale in Poland in 2018: sulfamethoxazole (up to 1358 kg/month), ofloxacin (up to 829 kg/month) and clindamycin (737 kg/month). Oxytetracycline, the total concentration of which was the highest and whose concentrations were very high also in the low-impacted water samples, as well as in one of the groundwater samples, is primarily used in veterinary medicine [20]. It is widely used as a feed additive in livestock husbandry, as a preventive measure and as a therapeutic agent [21]. It has been demonstrated that its widespread use not only pollutes sewage systems, but it results in their transport further to the aquatic environment via rainwater and surface runoff [21]. Moreover, tetracyclines along with fluoroquinolones are among the most persistent antimicrobial agents in the environment; thus, they tend to easily accumulate in aquatic and soil environments [7]. Erythromycin is another example of antibiotics that are widely used in both human and veterinary medicine. In veterinary medicine, it is used for the treatment of clinical and subclinical mastitis in cows, infectious diseases in cattle, sheep, swine and poultry and of chronic respiratory diseases in poultry [22]. Erythromycin has been reported as resistant to treatment by the STPs, resulting in its ubiquitous occurrence in final effluents [23]. Importantly, erythromycin can bind to biosolids and thus contaminate sewage sludge, which is widely used as an agricultural fertilizer. That way, this antibiotic can reach river water via surface runoff or reach groundwater [23]. Weiser et al. [24] suggested another three out of the most frequently detected antibiotics in this study, i.e., erythromycin, clindamycin and trimethoprim, to be among very commonly detected in the STP receiving water bodies and for this reason were recommended as of particular concern to the environment. The authors in [18] report that trimethoprim and sulfamethoxazole (which are very often used in a fixed combination) are among the most frequently detected antimicrobial agents in the European STP effluents with the detection frequency reaching around 90% of STP effluents. Among the reasons for such a situation may be their very long half-life values, of 20.3 days for sulfamethoxazole and <11.8 days for trimethoprim.

The concentrations and percentage share of antimicrobial agents varied significantly between the examined water samples. Out of the seven sites, only one was not contaminated with antimicrobial agents, and it was groundwater collected high in the Tatra Mountains (c.a. at 1600 m a.s.l.). However, in groundwater collected at the border of the Tatra National Park (at c.a. 1000 m a.s.l.), there were six antimicrobial agents detected with the total concentration of 27.29 ng/L. The three antibiotics with the highest concentrations in this GW sample comprised vancomycin, erythromycin and clindamycin. The high share of vancomycin was also observed in two other sites, considered as pristine (RW1) and low impact (RW2), where the total concentration of antimicrobial agents was 40.95 ng/L and 17.94 ng/L, respectively. Detection of this antibiotic both in surface water and in groundwater samples is not unprecedented. The authors in [20] reported the presence of this glycopeptide antibiotic in both influents and effluents of STPs as well as in the receiving river waters (both downstream and upstream of the examined STPs, suggesting that the

STP effluents are not the only sources of this antibiotic in the aquatic environment). One of the possible reasons for the detection of this antimicrobial agent in six out of the examined seven sampling sites, in relatively high share, is its relative resistance to degradation in conventional STPs (removal rate between 5 and 52%) [18]. The experiments carried out by Cao et al. [25] on the metabolism and degradation of vancomycin demonstrated that this antibiotic is not metabolized in the liver and as such it is entirely excreted in the urine in an unchanged form. Moreover, the half-life of vancomycin in water is approx. 9–10 days, which allows this antibiotic to persist and be transferred in the aquatic environment.

The mean and total concentrations of antibiotics increased dramatically by the discharge site from the STP (i.e., 9601.41 ng/L of all antimicrobials, the highest mean and maximum concentrations of most of the detected antibiotics and the highest number of antimicrobials detected). This observation is in line with the reports by other authors that the STP effluents play a major role in antimicrobials' input into the aquatic environment [4,18,20]. Moreover, while comparing the concentrations of antimicrobial agents in the sites upstream and downstream of the STP, it is also clear that apart from the STP, there might be other important sources of antibiotics in the aquatic environment. The total concentration of antibiotics in RW4 (c.a. 3 km downstream of the STP) is 1012.90 ng/L, while the total concentration in the RW5 site (c.a. 7 km downstream of the STP) is 2306.59 ng/L. Importantly, this site is characterized by the predominance of oxytetracycline, which is an important antimicrobial agent used not only in human but also in veterinary medicine. Its environmental input relies on, among others, surface runoff from fields fertilized with animal manure contaminated with this antimicrobial agent [21]. Generally, the evident differences in the concentrations of the examined antibiotics between the study sites affected their position in the cluster analysis shown in Figure 4. Pristine samples (GW1, GW2 and RW1) and one low impact sample (RW2) clearly stand out from the remaining sites, located downstream of the STP. A case study by Lenart-Boroń et al. [10] was a preliminary investigation pointing to the examined STP as the major contributor not only to the antibiotic contamination of the aquatic environment in the Białka river valley but also to the spread and dissemination of antibiotic-resistant bacteria and genetic determinants of some important resistance traits, such as extended-spectrum beta lactamase type of resistance among *E. coli*.

However, this is not only the case of the Białka river valley. Similar or even higher concentrations of antimicrobial agents in STP effluents and surface waters were observed by other authors. The authors of [20] examined two Polish STPs and their receiving surface water bodies (one in southern Poland, one in northern Poland) in terms of detection and quantification of 26 antimicrobial agents. Similarly, as in our study, the concentrations of antimicrobials detected in the examined samples varied within wide ranges. For instance, mean concentration of clindamycin in STP effluents ranged from 166 to 290 ng/L in the receiving river water upstream of the STP from 2.3 to 78 ng/L and downstream from 25.4 to 134 ng/L. For ciprofloxacin, the values were as follows: effluents of STPs—from 184 to 312 ng/L, river water upstream of the STPs—from 12 to 108 ng/L, downstream—from 95 to 182 ng/L [26]. In another European study, [27] identified both STPs and farming areas as the main emission sources of antibiotics in the Ebro river basin in Spain. Enrofloxacin and ciprofloxacin (fluoroquinolones) were the most frequently detected antibiotics, followed by sulfamethoxazole, sulfadiazine (sulfonamides) and trimethoprim (around 60% of samples). In STP effluents, azithromycin (macrolide) was detected in the highest concentrations, exceeding 5000 ng/L. The second antimicrobial with this respect was enrofloxacin (mean concentration of 1300 ng/L) and the third was sulfadiazine (300 ng/L). The effluents of slaughterhouses were characterized by high concentrations of azithromycin and trimethoprim. In a Portuguese case study, [4] observed ciprofloxacin to be most prevalent in groundwater (43% of samples), while clarithromycin, erythromycin and ciprofloxacin were the most prevalent in surface water samples (46 and 38% of samples). On the other hand, Hanna et al. [28] examined more than 200 aquatic samples in eastern China and found sulfapyridine, sulfamethoxazole, ciprofloxacin, enrofloxacin, levofloxacin, norfloxacin,

chloramphenicol, florfenicol, doxycycline and metronidazole at concentrations ranging from 0.3 to 3.9 ng/L in river water and from 1.3 to 12.5 ng/L in sewage.

As stated by numerous authors, consumption of antibacterials, which is very country-specific, is one of a few important factors affecting their presence in the aquatic environment [18,20,23]. In the considered period, the highest consumption was recorded for penicillins (for both human and veterinary medicine). This was followed by glycopeptide and MLS antibiotics in 2019 in human medicine and by other β-lactams and trimethoprim/sulfamethoxazole in 2020 [15]. In veterinary medicine, the second and third highest consumption were recorded for tetracyclines and the MLS group of antibiotics in both 2019 and 2020 [14]. This, however, is not reflected in the values of antibiotic detection in our study. Similarly, as in our study, [4] or [5] could not establish a clear link between the detection of antibiotics in STP effluents and their consumption. This was due to a number of reasons, including antibiotic metabolism in the human body, their chemical and biological stability, mobility in the environment and sewage treatment efficiency [4,5,29]. The authors in [18] explain rare detection of penicillins in the aquatic environment and in STP effluents by their chemical instability due to, e.g., physicochemical interactions (such as zinc and copper-mediated degradation), easy biodegradation (c.a. 90%) during STP treatment processes or widespread presence of β-lactamase producing bacteria in the environment. On the other hand, tetracyclines, which were detected in the highest concentrations in our study, are reported to be among the most persistent in the environment [7,30].

However, regardless of the examined discrepancies between the consumption vs. detection of antimicrobials, what needs to be emphasized at this point is that the detected concentrations of antibiotics are in most cases sublethal, and it can be considered that they may select bacterial resistance [31]. However, the antimicrobial resistance and its easy spread via the horizontal gene transfer [32] are not the only problems associated with environmental contamination by antibiotics. Antibiotics alter species richness and abundance of not only the target organisms but also the non-target ones, including other prokaryotes, algae, fish and zooplankton [30]. The effects can vary between the groups of antibiotics. For instance, the effects of the ones that are most frequently detected in our study, i.e., clindamycin, erythromycin and trimethoprim, on microbial communities were examined by [24]. The examined antibacterials increased biofilm thickness (affected biofilm exopolysaccharide composition) of the bacterial populations, decreased bacterial and algal biomass, decreased utilization of carbon sources and altered the numbers of protozoa and rotifers. Another highly frequent antibiotic in our study was ofloxacin, belonging to the class of fluoroquinolones. This group of antimicrobials is reported as highly persistent in the environment, thus easily exerting a chronic selective pressure on microbial communities [30] and being highly toxic to both prokaryotic and eukaryotic organisms [18]. Moreover, their genotoxicity and human health risks associated with their long-term use have been reported [33].

## 5. Conclusions

This study is one of only a few publications concerning the presence and concentrations of antimicrobial agents in Poland, and for this reason it adds important data to fill this huge knowledge gap. Out of 24 antibiotics monitored, we managed to detect 16 above the LOQ of the UHPLC/MS method. Among these, clindamycin was most frequently detected, and its concentration was the second highest among all antimicrobials. The highest concentration was observed for oxytetracycline, which is used only in veterinary medicine but for a wide range of purposes. A large variation in the antibiotic concentration was observed among the seven examined sites. No traces of antibiotics were detected in only one out of two groundwater samples (located c.a. at 1600 m a.s.l.). The second groundwater site as well as the river water sampling site located in the Tatra National Park were contaminated by antibiotics, with vancomycin having the greatest share among the antimicrobials detected in these sites. Not surprisingly, the STP proved to be the hotspot for dissemination of antibiotics in the aquatic environment in the Białka river valley. Comparing the antibiotic

consumption with the detection rates suggested that this parameter cannot be treated as the sole indicator of antibiotics spread in the environment. Judging by the results of this study, the rate of detection and concentrations of antibiotics in the Bialka river valley mostly reflect their stability in the environment. Finally, the antibiotics most frequently detected in the Białka river valley (clindamycin, erythromycin and trimethoprim) occur in sublethal concentrations that have the most dangerous impact on the aquatic environment. They exhibit a wide range of effects, including the stimulation of HGT or biofilm production, as well as toxicity to prokaryotic and eukaryotic organisms.

**Author Contributions:** Conceptualization, A.L.-B. and P.B.; methodology, J.P. and M.G.; software, P.B. and M.G.; validation, M.G., M.Ż., B.G. and P.B.; formal analysis, J.P.; investigation, J.P.; resources, A.L.-B., B.G. and M.G.; data curation, J.P.; writing—original draft preparation, A.L.-B.; writing—review and editing, P.B., J.P. and M.Ż.; visualization, A.L.-B. and P.B.; supervision, A.L.-B.; project administration, A.L.-B.; funding acquisition, A.L.-B., M.G., B.G. and M.Ż. All authors have read and agreed to the published version of the manuscript.

**Funding:** This research was funded by the measures of the National Science Center in Poland, under the project MINIATURA, no. 2018/02/X/NZ9/00867, by the statutory measures of the University of Agriculture in Kraków within grant no. Sub 010014-D011, by the statutory measures of the Jerzy Haber Institute of Catalysis and Surface Chemistry, Polish Academy of Sciences and Jagiellonian University (grant no. K/KDU/000297; K/KDU/000618).

**Institutional Review Board Statement:** Not applicable.

**Informed Consent Statement:** Not applicable.

**Data Availability Statement:** The authors declare that the data supporting the findings of this study are available within the article.

**Acknowledgments:** Justyna Prajsnar acknowledges the fellowship with the project POWR.03.02.00-00-I013/16. We would like to acknowledge Martin Mullet from the Centre for Ecosystems, Society and Biosecurity, Forestry Commission UK, for his help in preparing this article in correct English language.

**Conflicts of Interest:** The authors declare no conflict of interest.

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
