# Peer review of "Antibiotics in Groundwater and River Water of Białka—A Pristine Mountain River"

_applsci, doi:10.3390/app122412743_

Round 1
Reviewer 1 Report
I accept this manuscript in its present form
Author Response
Thank you very much for the positive comment and accepting the manuscript.
Reviewer 2 Report
Explain why the recovery is low? (SPE recovery ranged from 9.94% to c.a. 100% 150 [10]. )
Author Response
Thank you for the positive assessment of our manuscript.
Please find below the response to the question:
Explain why the recovery is low? (SPE recovery ranged from 9.94% to c.a. 100% 150 [10]. )
Response: the SPE recovery was determined for all antimicrobial agents assessed in the course of the study, based on three replicates. The detailed values are presented in the supplementary data of the cited paper (published by our team in 2020). For some antibiotics the SPE recovery was low (e.g. 9.94% for amoxicillin and 17.50% for ampicillin), due to their lower affinity to the SPE cartridge. However, the final SPE recovery rate was used to calculate the final concentration of antibiotics in the examined samples, as also stated in the cited paper. We added this information to the section of Materials and Methods that refers to the SPE recovery rates.
Reviewer 3 Report
Questions:
Line 102: Has it been verified that ATBs are not adsorbed on the PP bottle?
Figure 1: Why was the groundwater sample near RW2-5 not included?
Line 114-120: On what basis was the observed ATB selected?
Figure 5: How do you explain the big difference in penicillin ATB consumption and their low occurrence in the samples?
Line 322-323: Can the source (tourists vs. Veterinary use) be inferred from the detected ATBs in GW2?
Have metabolites also been analyzed for some ATBs?
Author Response
Dear Reviewer,
Thank you very much for positive comments of our manuscript and the questions.
Below you can find our responses to the questions:
1. Line 102: Has it been verified that ATBs are not adsorbed on the PP bottle?
No, it has not been verified in this study. However, we strongly believe that the use of PP bottles did not affect the resulting concentrations of the examined antimicrobial agents. Firstly, in a number of our studies we used a few different types of storage materials (e.g. PET, PP or glass) and we have not observed any distinct differences in the antibiotic concentrations within the same samples. Secondly, as stated in the materials and methods (l. 104), the samples were mostly processed immediately after being transported to the laboratory (which did not take longer than 2-3 hours). Otherwise, they were frozen at -20°C, which prevents the antibiotics from degradation or other form of behavior that affects their concentrations. The third argument that supports our statement that the examined antibiotics have not been adsorbed on PP bottles is that we have searched the literature on this topic and found the following information:
Atugoda et al. (2021) Interactions between microplastics, pharmaceuticals and personal care products: Implications for vector transport, Environment International 149 (2021) 106367
or
Yu et al. (2022) Selective adsorption of antibiotics on aged microplastics originating from mariculture benefits the colonization of opportunistic pathogenic bacteria. Environmental Pollution 313 (2022) 120157
state that the concentration of antibiotics on microplastics ranged from no detected to 25.66 ng/g (microplastic). The only antimicrobial agent that was reported to be adsorbed on microplastics, and included in our analysis, is tetracycline. However, these reports refer to microplastics that are characterized by significantly larger surface that the PP bottle.
Also:
Loeuille, G.; D’Huart, E.; Vigneron, J.; Nisse, Y.-E.; Beiler, B.; Polo, C.; Ayari, G.; Sacrez, M.; Demoré, B.; Charmillon, A. Stability Studies of 16 Antibiotics for Continuous Infusion in Intensive Care Units and for Performing Outpatient Parenteral Antimicrobial Therapy. Antibiotics 2022, 11, 458. https://doi.org/10.3390/antibiotics11040458
examined the stability of antibiotics in various conditions, including in polypropylene syringes and their results suggested that the shortest stability was reported for meropenem (4 hours), followed by cefotaxime (6 hours), to even 48 hours (e.g. vancomycin) and these results were obtained for room temperature or 37 degrees C. None of our samples were stored for that long.
2. Figure 1: Why was the groundwater sample near RW2-5 not included?
Response: the area around the sampling sites RW2-5 is urbanized, in adjacent to the center of the Białka Tatrzańska village. This results in a variety of anthropogenic factors affecting the groundwater that could be collected there. Thus, to avoid extra uncertainty (these factors were not fully recognized), we decided not to include any groundwater collected from sites like e.g. private wells. There is also no conveniently located piezometer that would allow to collect groundwater in the systematically controlled manner. Therefore, as stated in the Materials and Methods / Sample collection section (l. 96), the groundwater sampling sites were located in the Tatra National Park, which is assumed to be a pristine region. The borders of the Tatra National Park do not reach further downstream of the Białka river (the area of RW2-5 sites).
3. Line 114-120: On what basis was the observed ATB selected?
In order to select the antimicrobial agents for the study, we first screened the antimicrobial agents most widely used in human and veterinary medicine (as stated in the Materials and Methods – l. 110-111) using the qualitative approach – based on water and methanol extracts of antimicrobial susceptibility test disks. Based on the antimicrobial agents that first gave the positive results in qualitative assessment, we selected these that gave the most promising results (i.e. exceeded the LOD and LOQ baselines) and then adopted the quantitative approach using the pure standards.
4. Figure 5: How do you explain the big difference in penicillin ATB consumption and their low occurrence in the samples?
Response: the significant difference between consumption and detection of penicillins in water samples is due to their instability in the environmental conditions, as reported by e.g. Felis et al. (2020) – cited as [18]. This has also been explained by us in the Discussion section (l. 397-403).
5. Line 322-323: Can the source (tourists vs. Veterinary use) be inferred from the detected ATBs in GW2?
Response: It will be difficult to verify the source of the antimicrobial agents detected in water in general. It will be possible only in the case of antimicrobial agents that are solely used in veterinary medicine, such as oxytetracycline, which is not used in veterinary medicine. The antibiotics detected in GW2 include erythromycin, ofloxacin, clindamycin, vancomycin , trimethoprim and sulfamethoxazole. All these antimicrobial agents are used in both human and veterinary medicine. In the case of human medicine – the source can also vary – it may not only be due to the tourists present in the region, but also as a result of general frequent use of antimicrobial agents in medicine and their circulation in aquatic environment. One of the towns located nearby Tatra National Park is Zakopane, which is not only frequently visited by tourists, but also there are nearly 30,000 citizens of this town and there are three hospitals.
6. Have metabolites also been analyzed for some ATBs?
Response: No, the metabolites of antibiotics have not been analyzed in this study. We plan to undertake this type of analysis but this is a future direction.